# Comprehensive transcriptomic analysis of cell lines as models of primary tumors across 22 tumor types

K. Yu [1,2], B. Chen [3,4], D. Aran [1], J. Charalel[5], C. Yau[6,7], D.M. Wolf[7], L.J. van 't Veer [8], A.J. Butte [1,2], T. Goldstein[1] & M. Sirota[1,2]

Cancer cell lines are a cornerstone of cancer research but previous studies have shown that not all cell lines are equal in their ability to model primary tumors. Here we present a comprehensive pan-cancer analysis utilizing transcriptomic profiles from The Cancer Genome Atlas and the Cancer Cell Line Encyclopedia to evaluate cell lines as models of primary tumors across 22 tumor types. We perform correlation analysis and gene set enrichment analysis to understand the differences between cell lines and primary tumors. Additionally, we classify cell lines into tumor subtypes in 9 tumor types. We present our pancreatic cancer results as a case study and find that the commonly used cell line MIA PaCa-2 is transcriptionally unrepresentative of primary pancreatic adenocarcinomas. Lastly, we propose a new cell line panel, the TCGA-110-CL, for pan-cancer studies. This study provides a resource to help researchers select more representative cell line models.

[1] Bakar Computational Health Sciences Institute, University of California San Francisco, San Francisco 94158 CA, USA. [2] Department of Pediatrics, University of California, San Francisco, San Francisco 94143 CA, USA. [3] Department of Pediatrics and Human Development, College of Human Medicine, Michigan State University, Grand Rapids 49503 MI, USA. [4] Department of Pharmacology and Toxicology, College of Human Medicine, Michigan State University, Grand Rapids 49503 MI, USA. [5] Department of Genetics, Stanford University, Stanford 94305 CA, USA. [6] Buck Institute for Research on Aging, Novato 94945 CA, USA. [7] Department of Surgery, University of California, San Francisco, San Francisco 94143 CA, USA. [8] Department of Laboratory Medicine, University of California, San Francisco, San Francisco 94143 CA, USA. Correspondence and requests for materials should be addressed to M.S. (email: Marina.Sirota@ucsf.edu)

Cancer cell lines are an integral part of cancer research and are routinely used to study cancer biology and to screen anti-tumor compounds. While they are relatively inexpensive and easy to grow under laboratory conditions, cell lines have known limitations as preclinical models of cancer and many promising candidate drug compounds have failed to show utility among patient populations[1,2]. Prior studies in ovarian cancer[3], liver cancer[4], and breast cancer[5,6] have shown that cell lines differ in their ability to represent the primary tumors they were derived from, suggesting that using more appropriate cell lines for cancer studies may increase the translatability of preclinical findings. While these previous studies are valuable resources for researchers studying select tumor types, there is a need for a comprehensive pan-cancer analysis of cell lines and primary tumors.

The generation of large public molecular data sets has allowed researchers to investigate cancer biology at a scale that was unheard of a decade ago. In particular, The Cancer Genome Atlas (TCGA)[7] research group has collected and characterized the molecular profiles of tumors from over 11,000 patients across 33 different tumor types. They provide clinical, transcriptomic, methylation, copy number, mutation, and proteomic data to facilitate the in-depth interrogation of cancer biology at multiple molecular and clinical levels. In addition, the Broad Institute's Cancer Cell Line Encyclopedia[8] is another large-scale research effort which characterized over 1000 human-derived cancer cell lines across 36 tumor types and provides transcriptomic, copy number, and mutation data.

Previous studies have integrated data from both of these data sets to evaluate cell lines as models of specific tumor types. For example, Domcke et al. focused primarily on copy number alterations and mutation data to evaluate cell lines as models of high-grade serous ovarian carcinomas (HGSOC)[3]. They created a cell line suitability score using features of HGSOC and discovered that the most commonly used cell lines do not seem to resemble HGSOC tumors, and the cell lines most representative of HGSOC have very few publications. Similarly, Chen et al. compared hepatocellular carcinoma primary tumor samples to cell lines using transcriptomic data and found that nearly half of the hepatocellular carcinoma cell lines in CCLE do not resemble their primary tumors[4]. In breast cancer, Jiang et al. compared gene expression, copy number alterations, mutations, and protein expression between cell lines and primary tumor samples[5]. They created another cell line suitability score by summing the correlations across all four molecular profiles, although it is notable that only gene expression and copy number alterations had a substantial effect on their score as mutations and protein expression had extremely low correlations across all cell lines ($R < 0.1$). In another breast cancer study, Vincent et al. compared transcriptomic data between cell lines and primary tumor samples and identified basal and luminal cell lines that were most similar to their respective breast cancer subtypes[6]. While these studies provide insight into specific tumor types, here we hope to provide researchers with a pan-cancer resource that is, to the best of our knowledge, the most comprehensive to date. In addition, unlike previous studies, we adjust for tumor purity which can be a significant confounder in primary tumor transcriptomic data[9].

Cancer is an incredibly heterogeneous disease that can often be stratified into clinically relevant subtypes with different prognosis and responses to treatments. While specific genomic alterations or histological markers have been used to stratify tumors, gene expression is commonly used to group tumors into molecular subtypes[10–12]. Breast cancers, for example, can be divided into five intrinsic molecular subtypes based on gene expression profiles with distinct clinical outcomes[13]. While much progress has been made in separating primary tumors into biologically distinct subtypes, few publications have attempted to apply these subtypes to cell line models. Our study seeks to provide subtype classifications for cell lines to aid researchers interested in subtype-specific studies or drug screens.

The National Cancer Institute's NCI-60 cell lines are perhaps the most well-studied human cancer cell lines, and have been in use for nearly three decades by both academic and industrial institutions for drug discovery and cancer biology research[14]. The NCI-60 panel contains 60 human tumor cell lines representing nine human tumor types: leukemia, colon, lung, central nervous system, renal, melanoma, ovarian, breast, and prostate. Over 100,000 antitumor compounds have been screened using this cell line panel, generating the largest cancer pharmacology database worldwide. While this cell line panel has provided valuable insight into mechanisms of drug response and cancer biology, new large public molecular data sets allow us to compare the NCI-60 cell lines to primary tumor samples and propose more representative cell lines for an improved cancer cell line panel.

In this study, we compared transcriptomic profiles from cell lines and primary tumor samples across the 22 tumor types covered by both TCGA and CCLE. We observed the confounding effect of primary tumor sample purity in our analysis, and we adjusted for purity in our correlation analysis and differential expression analysis of cell lines and primary tumor samples. We found that cell-cycle-related pathways are consistently upregulated in cell lines, while immune pathways are consistently upregulated across the primary tumor samples. Next, we classified cell lines into subtypes across nine tumor types. We then present our analysis of pancreatic adenocarcinoma (PAAD) cell lines and primary tumor samples and show that we are able to identify a cell line that originated from a different cell-type lineage compared with the primary tumor samples. Although only our PAAD analysis is presented in the main text, we also analyzed the other 21 tumor types and present our results as a web application and a resource to the cancer research community (http://comphealth.ucsf.edu/TCGA110CL). Last, we selected the cell lines that were the most correlated to their primary tumor samples across 22 tumor types and propose a new cell line panel, the TCGA-110-CL, as a more appropriate and comprehensive panel for pan-cancer studies.

## Results

**Pan-cancer comparison of expression profiles**. We compared RNA-seq profiles from 8282 primary tumors from TCGA with 666 cell lines from CCLE across 22 overlapping tumor types (Supplementary Data 1, 2). Primary tumors were used in all tumor types except for SKCM, in which case the metastatic tumors were included because the SKCM TCGA cohort was primarily focused on metastatic tumors. We normalized counts using the upper-quartile method and corrected for batch effects related to different sequencing platforms using ComBat[15] (Supplementary Fig. 1). For each tumor type, we then adjusted for tumor purity in the primary tumor samples and calculated correlation coefficients between primary tumor samples and cell lines using the 5000 most variable genes, as these genes are the most likely to be biologically informative (see the Methods section). To understand the biological processes captured by the 5000 most variable genes, we performed gene ontology analysis on the top 10% of genes driving the correlations in each tumor type and found that many developmental pathways were enriched (Supplementary Data 3). This is consistent with the view that developmental pathways are often altered in cancer[16–18]. A full matrix of the cell line and primary tumor correlations are provided in Supplementary Data 4.

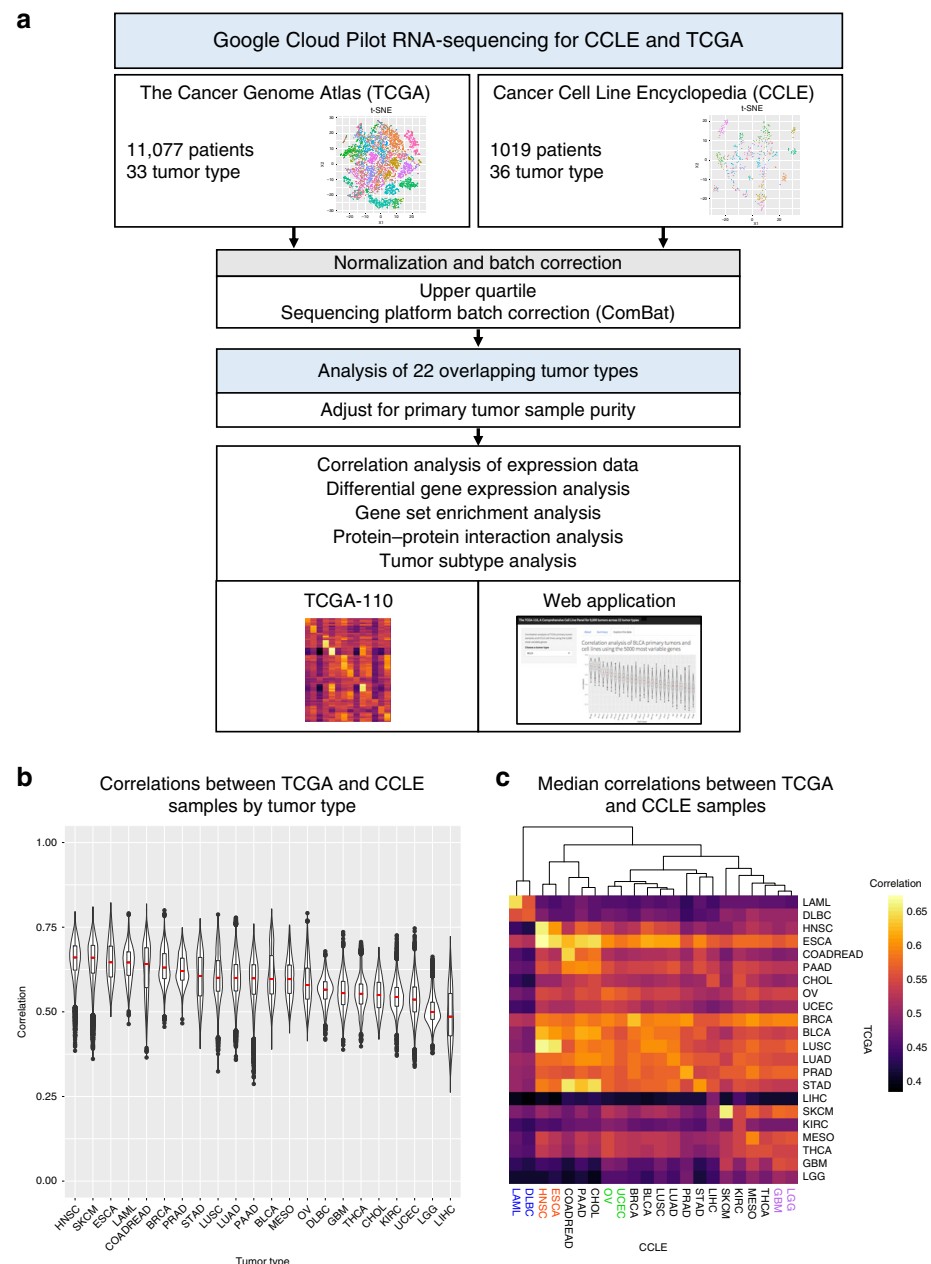

**Fig. 1** Pan-cancer analysis of cell lines and matching primary tumor samples. **a** Study design. RNA-seq data were downloaded from the Google Cloud Pilot RNA-sequencing for CCLE and TCGA project for 22 cancer types that overlapped between the CCLE and TCGA data sets. The data were normalized, batch corrected, and adjusted for tumor purity during the analysis. **b** Correlation analysis of the CCLE and TCGA data. Each sample in the violin plot corresponds to the Spearman correlation between one cell line and one primary tumor sample using the 5000 most variable genes. In the overlaid boxplot, the red center line depicts the median, the box limits depict the upper and lower quartiles, and the whiskers depict 1.5 times the interquartile range. **c** Heatmap of median correlations between all tumor types in CCLE compared with all tumor types in TCGA

The median correlation coefficients between cell lines and their matched tumor samples were relatively consistent across tumor types, from 0.66 in head and neck squamous cell carcinoma (HNSC) to 0.49 in liver hepatocellular carcinoma (Fig. 1b). Within tumor types, the correlation coefficient ranges were largest in PAAD (0.29–0.76), LUSC (0.32–0.79), and LIHC (0.26–0.72), which likely reflect the amount of heterogeneity within each tumor type and suggest that some primary tumor samples are well matched with cell lines, while others may lack representative cell line models.

Our clustering analysis of cell line and primary tumors correlation coefficients largely captures known biological relationships between the tumor types (Fig. 1c). The first split in our clustering analysis depicts the large difference between hematopoietic tumor types and solid tumor types, previously shown in other studies[3]. Within the solid tumor cluster, tumor types from similar cell of origin generally clustered together, such as ovarian serous cystadenocarcinoma (OV) and uterine corpus endometrial carcinoma (UCEC), glioblastoma (GBM) and lower grade glioma (LGG), and esophageal carcinoma (ESCA) and HNSC. Interestingly, we observe that sometimes the highest correlation coefficients are not necessarily between cell line and primary tumor samples from the same tumor type. In fact, in 8/22 tumor types, primary tumor samples have higher correlation

coefficients with other tumor cell lines than their own. These tumor types are BLCA (highest correlation with HNSC), CHOL (highest correlation with LIHC), ESCA (highest correlation with HNSC), LGG (highest correlation with GBM), STAD (highest correlation with COADREAD), LUSC (highest correlation with HNSC), LUAD (highest correlation with PAAD), and UCEC (highest correlation with OV). While this may indicate poor differentiation in the cell lines or primary tumor sample or lack of appropriate cell line models, many of these tumor types have higher correlations with a related tumor type (e.g., LGG and GBM, STAD and COADREAD, UCEC and OV, ESCA and HNSC).

To verify that the results of our transcriptomic-based correlation approach were consistent with previous publications, we compared our cell line rankings for OV to the cell line rankings in Domcke et al. which evaluated high-grade ovarian cancer cell lines based on copy number alterations and selected mutations (Supplementary Data 5)[3]. Our results were highly correlated (Spearman's rho = 0.59, p-value = 5.837e-05), which suggests that our cell line rankings capture much of the same information as more curated ranking methods that use genomic alterations.

**Tumor purity drives primary tumor and cell line differences**. To explore the differences between cell lines and primary tumor samples, we initially performed our correlation and differential gene expression analysis across all 22 tumor types, without accounting for tumor purity of the primary tumor samples (Fig. 2a). In our correlation analysis, we compared the cell line correlations with primary tumor samples in the top quartile of tumor purity to the cell line correlations with primary tumor samples in the bottom quartile of tumor purity for the 20 solid tumor types for which we have tumor purity information (Fig. 2a). In 75% (15/20) of these tumor types, the cell lines were significantly more correlated with primary tumor samples in the top quartile of purity compared with the primary tumor samples in the bottom quartile of purity, suggesting that the individual correlation coefficients are reflecting, to a certain extent, the amount of non-tumor cells present in the primary tumor samples. Similarly, we found a significant positive relationship (R = 0.17, p-value < 2.2e-16) between primary tumor sample purity and the cell line-primary tumor correlation coefficients, suggesting that tumor purity is a confounder in our correlation analysis. Furthermore, when we performed Gene Set Enrichment Analysis (GSEA) on the differential expression results using the hallmark gene sets from the MSigDB Collections[19] and the hallmarks of cancer pathways[20], we saw that the gene sets involved in immune processes are consistently upregulated in primary tumor samples, suggesting that the largest biological signal from the TCGA samples can likely be attributed to the immune cell infiltrate that are present in the primary tumor samples and absent in the pure cell line populations (Supplementary Fig. 2C).

After adjusting for primary tumor sample purity in our correlation analysis, we confirmed that there was no longer a significant positive relationship between primary tumor sample purity and cell line-primary tumor correlation coefficients (R = −0.02, p-value < 2.2e-16). In addition, we found that only one tumor type (LGG) retained significantly higher correlations between cell lines and the primary tumor samples in the top quartile of purity compared with cell lines and primary tumor samples in the bottom quartile of purity (Supplementary Fig. 2D). We then performed differential expression analysis using tumor purity as a covariate to explore differences in cancer cell biology, while minimizing the influence of tumor infiltrating cells. The

number of differentially expressed genes ranged from 1157 in esophageal carcinoma (ESCA) to 4076 in low-grade glioma (LGG) (Supplementary Table 1). We identified 87 genes that were upregulated in primary tumor samples across 20 of the tumor types analyzed, and we found a significant number of interactions among these genes (PPI enrichment p-value < 1.0e-16) (Fig. 2b). This PPI network was enriched for genes in the immune response pathway (false discovery rate = 5.51e-06), suggesting that we were not fully able to remove the contribution of the immune infiltrate. However, the GSEA results show a much weaker enrichment of immunological pathways upregulated in the primary tumor samples (Fig. 2c, d).

No individual genes were significantly upregulated in cell lines across 90% of the tumor types analyzed. However, gene sets involved in cell-cycle progression (e.g., E2F targets, G2M checkpoint, and Myc targets) and genome instability were significantly enriched in cell lines in our GSEA of MSigDB Hallmark Gene Sets and the Hallmarks of Cancer pathways (Fig. 2c, d). These results demonstrate how GSEA can be more informative than analyzing individual upregulated genes alone. In addition, the enrichment of proliferative gene sets in cell lines across the tumor types suggests a common response to in vitro culturing conditions.

**Predicting subtypes in cancer cell lines**. In order to predict the subtype of individual cancer cell lines, we applied the Broad Institute's Nearest Template Prediction (NTP) method[21], which has previously been used to predict the subtypes of cancer cell lines[22]. Briefly, this method involves generating gene templates for each subtype by identifying genes that are upregulated in each subtype compared with the other subtypes. The distances between the sample to be classified and each subtype template is then calculated, and the sample is predicted to belong to the subtype with the smallest template distance (Fig. 3a).

Like Sveen et. al, we modified this method to create a classifier that can be applied to cancer cell lines after training the classifier on primary tumor samples[22]. We began with the 18 TCGA tumor types for which we had subtype information from TCGA publications[23–39], and randomly divided these samples into training sets (80%) and test sets (20%). After generating our initial subtype templates using the training set of primary tumor samples, we removed genes that are differentially expressed between primary tumors and cell lines as we wanted to enrich our subtype templates for genes that are consistent between primary tumors and cell line models. We also filtered out genes that are not highly expressed in at least a subset of the cell lines, as we wanted to retain genes that are robust and informative in cell lines. This filtering step can also enrich for cancer-intrinsic genes, since cell lines are pure populations of cancer cells. To verify that our classifier is still able to predict tumor subtypes after enriching for cell line relevant genes, we applied the classifier to the test set of held out primary tumor samples. Overall, 9/18 tumor types had a classification accuracy greater than or equal to 80% in the test set. We then applied the classifiers of these nine tumor types to their respective cell lines and predicted the subtypes of the individual cell lines (Fig. 3b). While all the primary tumor subtypes are predicted to be present in their respective cell lines, the proportions of subtypes significantly differ between primary tumors and cell lines in BRCA (chi-squared p-value < 2.2e-16), LUAD (chi-squared p-value = 9.5e-4), and SKCM (chi-squared p-value = 4.7e-5). This is likely because certain tumor subtypes have a higher rate of cell line generation than others due to their biology. We present the results of our cell line subtype predictions in Supplementary Data 6. We also include the ranks of each cell line compared with the primary tumors of the individual subtypes.

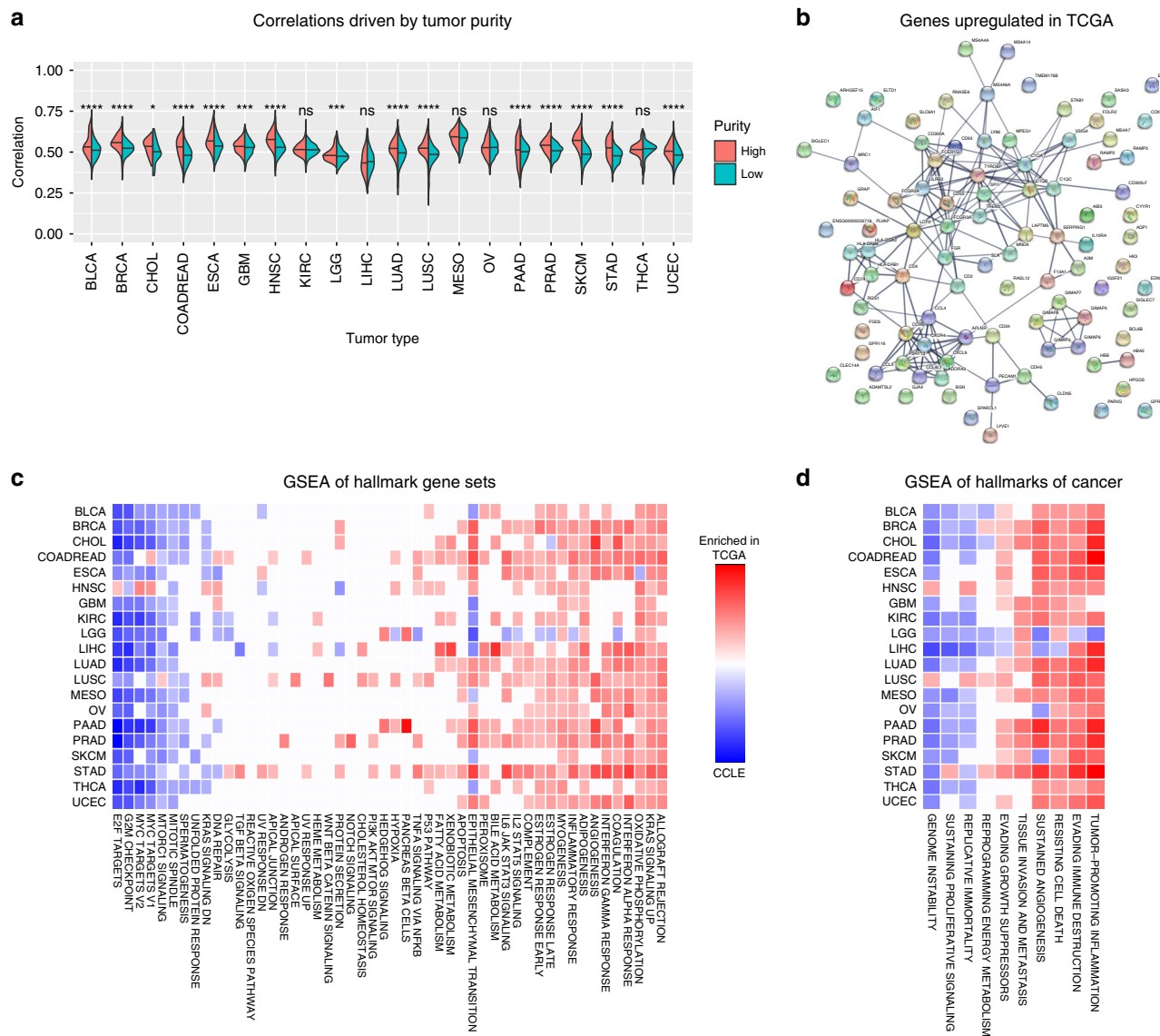

**Fig. 2** Primary tumor sample/cell line correlations driven by tumor purity. **a** Correlations between cell lines and high purity primary tumor samples (red) are significantly higher than correlations between cell lines and low purity primary tumor samples (turquoise) in 15/20 tumor types using the one-sided Wilcoxon test. *P*-values are indicated by symbols above the violin plots with ns corresponding to *p*-value > 0.05, one star corresponding to *p*-value < = 0.05, two stars corresponding to *p*-value < = 0.01, three stars corresponding to *p*-value < = 0.001, and four stars corresponding to *p*-value < = 0.0001. The median correlation coefficients are depicted by the horizontal black lines in the violin plots. **b** STRING analysis of protein–protein interactions for the 95 genes upregulated in primary tumor samples in all 20 of the analyzed tumor types (PPI enrichment *p*-value < 1.0e-16). Line thickness denotes confidence of the interaction, and only high confidence interactions are shown. The PPI network is enriched for immune response pathway genes (false discovery rate = 5.51e-06). **c** Gene Set Enrichment Analysis (GSEA) between primary tumor samples and cell lines using hallmark gene sets from MSigDB. NES are shown for pathways with FDR < 5%. Blue boxes indicate enrichment in cell lines, and red boxes indicate enrichment in primary tumor samples. Gene sets related to cell-cycle progression are enriched in cell lines across tumor types, and immune pathways are enriched in primary tumors. **d** GSEA of hallmarks of cancer pathways. Genome instability is enriched in cell lines across all tumor types, and tumor-promoting inflammation is enriched in primary tumors

**Case study: evaluating PAAD cell lines**. PAAD is often diagnosed at an advanced stage, and is predicted to become the second leading cause of cancer mortality by the year 2030. PAAD tumors can be divided into basal or classical molecular subtypes, with significantly lower survival associated with the basal subtype[40]. We utilize these subtypes in our study of PAAD presented here. While only the analysis for PAAD is shown, analysis of the other tumor types are available in our web application (http://comphealth.ucsf.edu/TCGA110CL). For each tumor type, we adjusted for primary tumor purity and compared the expression profiles of the primary tumor samples to the 932 cell line expression profiles in a correlation analysis. We included

tumor subtype predictions for the nine tumor types where the prediction accuracy in the test set was greater ≥80%.

We compared the correlations between PAAD primary tumor samples and all 932 cell lines grouped by cell line tissue of origin (Fig. 4a). The PAAD primary tumor samples are most correlated with cell lines originating from the pancreas, which contains all the PAAD cell lines. The correlation coefficients between PAAD primary tumor samples and cell lines from the pancreas; however, are not significantly higher than the correlation coefficients between PAAD primary tumor samples and cell lines from the second most correlated tissue of origin, the biliary tract. This suggests that pancreatic cell lines and

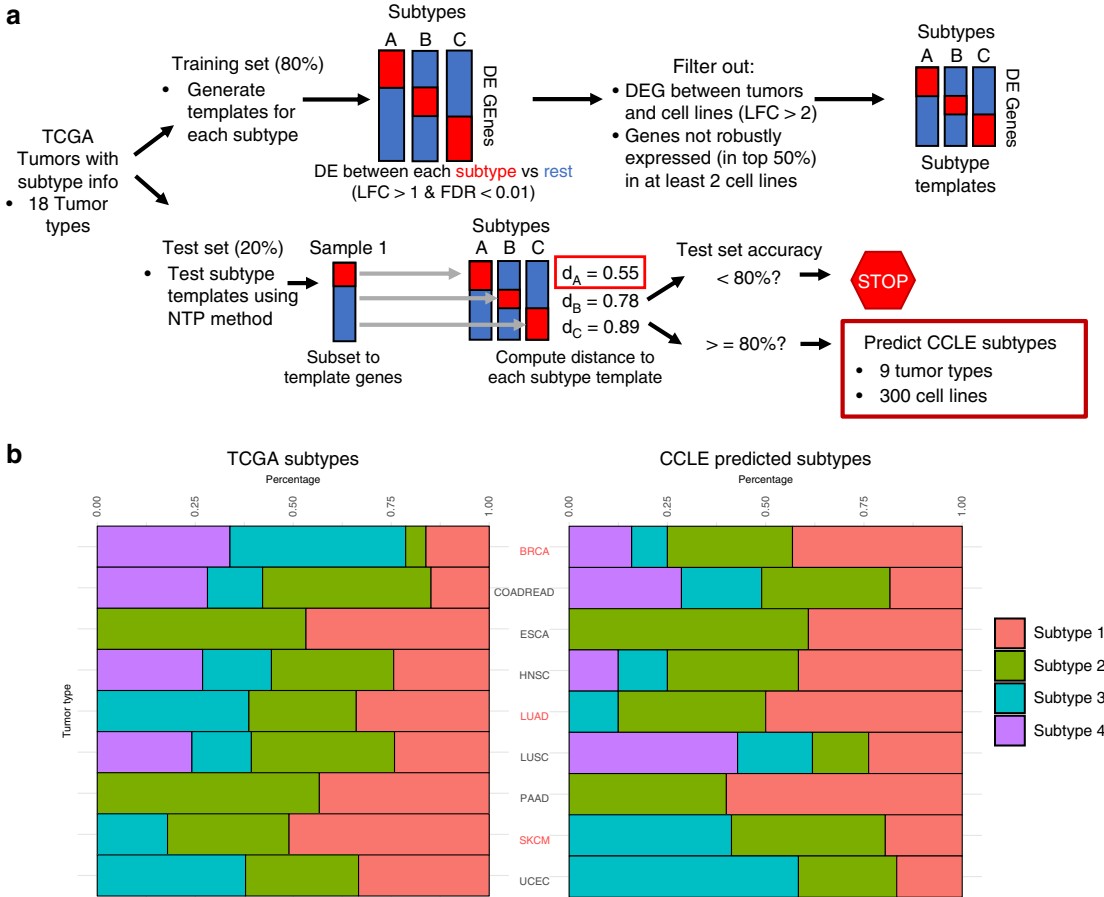

**Fig. 3** Cell line tumor subtype predictions. **a** Overview of the tumor subtype prediction method used in the study. TCGA tumors are divided into a training (80%) set to identify genes that are upregulated in each tumor subtype compared with the other tumor subtypes (LFC > 1, FDR < 0.01). Subtype templates are then filtered to remove genes that are differentially expressed between primary tumor samples and cell lines (LFC > 2) and genes that are not robustly expressed in at least two cell lines to generate cell line relevant subtype templates. These subtypes of the TCGA test set (20%) are then predicted using the Nearest Template Prediction method and if classification accuracy is greater than 80%, the gene templates are then applied to the CCLE cell lines to predict the cell line subtypes. **b** The proportion of tumor subtypes within the TCGA cohort (left) and the predicted tumor subtypes in the CCLE cell lines (right) for tumor types with prediction accuracy greater than 80%. The tumor types highlighted in red (BRCA, LUAD, SKCM) have significantly different proportions of subtypes when comparing the TCGA subtypes to the CCLE predicted subtypes

biliary tract cell lines share a large amount of biology, perhaps because of their ductal nature or close anatomical proximity. We next compared individual PAAD cell lines to the PAAD primary tumor samples (Fig. 4b). The median correlation coefficients of the cell lines ranged from 0.67 to 0.49, suggesting that some cell lines are less suitable as models of primary tumor samples than others. Within the cell lines, however, the standard deviations of the correlation coefficients are relatively low (0.08–0.03). This suggests that between cell line differences are larger than within cell line differences, the latter of which reflects the variability of the primary tumor samples. Interestingly, we found that the cell line with the second lowest median correlation, QGP1, is derived from a pancreatic neuroendocrine tumor rather than a PAAD, and the cell line with the lowest correlation, MIA PaCa-2, was derived from an adenocarcinoma but has been shown to also express neuroendocrine differentiation[41]. This suggests that our correlation approach is able to distinguish between cell lines derived from different cell types or cell lines that may not be representative of PAAD. Of potential concern, the cell line with the lowest median correlation coefficient, MIA PaCa-2, is commonly used as an adenocarcinoma cell line model and has over 1000 PubMed citations.

Next, we incorporated primary tumor subtype information from Moffit et al., which classified the PAAD primary tumor samples into two molecular subtypes[40]. We did not see strong clustering by primary tumor subtypes in our primary tumor/cell line correlation matrix (Fig. 4c). This suggests that our correlation approach using the 5000 most variable genes, while useful in showing global differences between cell lines and primary tumor samples, may not be adequate for distinguishing between specific tumor subtypes.

We then used the Nearest Template Prediction method to predict the subtypes of the pancreatic cancer cell lines (Fig. 4d). After deriving the subtype template genes from a training set (80%) of the PAAD TCGA tumors and applying our filtering criteria, we tested these subtype templates on a test set (20%) of held out PAAD TCGA tumors. We achieved a classification accuracy of 96% (Fig. 4d, top), suggesting that our classifier is able to successfully predict pancreatic subtypes even after applying our filtering criteria to enrich for cell line-relevant genes. We then used our classifier to predict the subtypes of the PAAD cell lines. In all, 15 cell lines were predicted to belong to the basal subtype, 10 cell lines were predicted to belong to the classical subtype, and 16 cell lines had an FDR > 0.05 and could not be assigned a subtype (Fig. 3d, bottom). 15 PAAD cell lines in our study were

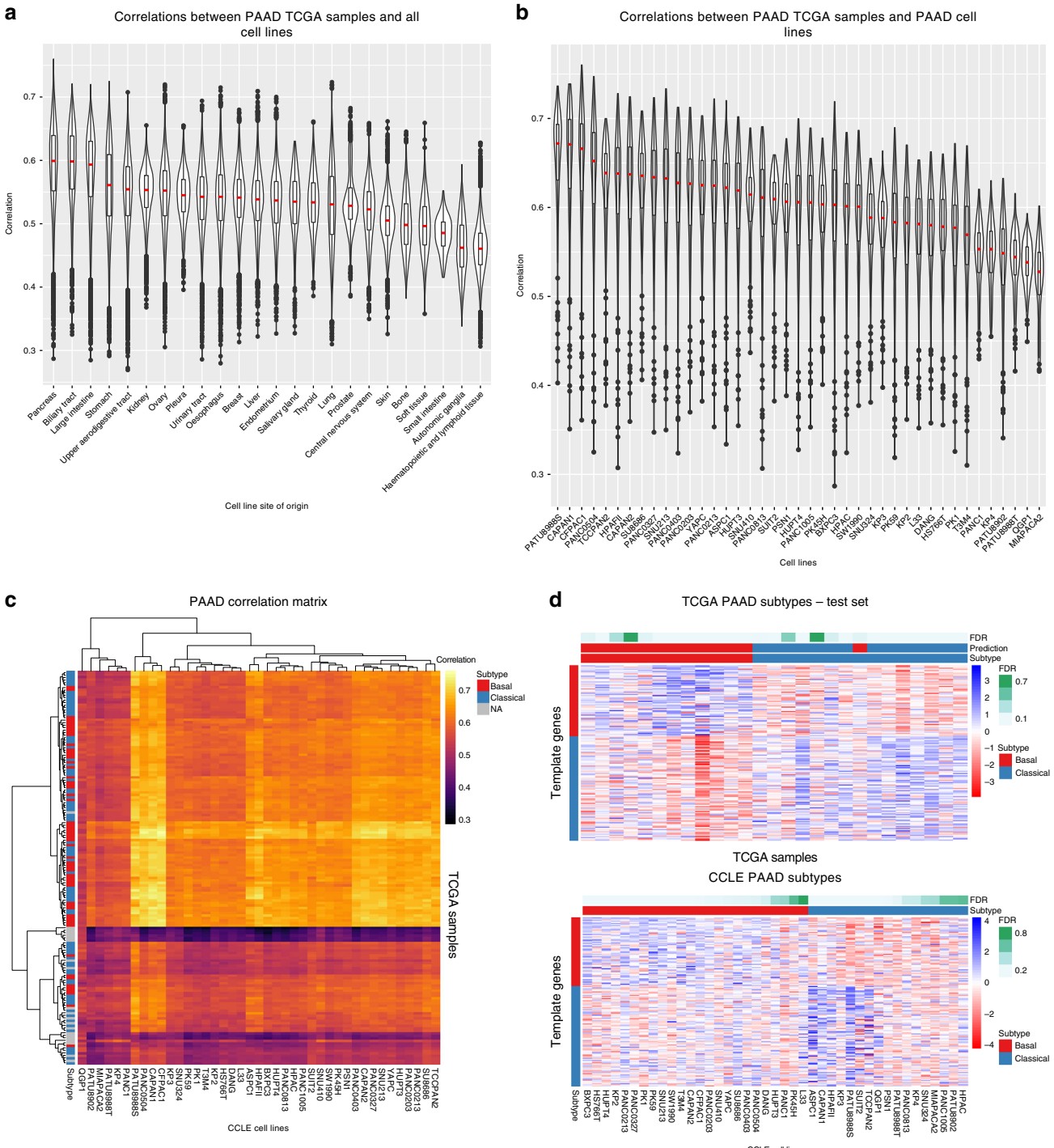

**Fig. 4** Correlation analysis of pancreatic adenocarcinoma (PAAD) tumor samples and cell lines. **a** Violin plot of Spearman's correlations between primary PAAD samples and all CCLE cell lines using 5000 most variable genes. The correlations are separated by cell line tissue of origin (*x*-axis). In the overlaid boxplot, the red center line depicts the median, the box limits depict the upper and lower quartiles, and the whiskers depict 1.5 times the interquartile range (IQR). Primary pancreatic tumor samples are most correlated with cell lines from the pancreas followed by the biliary tract. **b** Violin plot of Spearman's correlations between PAAD cell lines and PAAD tumor samples, separated by cell lines (*x*-axis). The median correlation coefficients, which are depicted by the red lines, range from 0.67 to 0.49. In the overlaid boxplot, the box limits depict the upper and lower quartiles, and the whiskers depict 1.5 times the IQR. **c** Heatmap showing the Spearman's correlations between PAAD cell lines (*x*-axis) and PAAD primary tumor samples (*y*-axis). The color bar on the *y*-axis indicates the subtype of the TCGA primary tumor samples. **d** Heatmaps show the expression levels of the PDAC template genes for the basal and classical PAAD subtypes. Top graph shows the TCGA PAAD test set with annotation color bars showing actual subtype, predicted subtype, and FDR for the subtype predictions. Bottom graph shows PAAD cell lines with annotation color bars showing predicted subtype and FDR for the subtype predictions

also analyzed by the Moffit et al. publication[40]. Out of these 15 cell lines, 10 cell lines passed our subtype prediction FDR cutoff of 0.05. While the Moffit et al. publication predicted all ten of these cell lines to belong to the basal subtype, we predicted that eight of these cell lines belong to the basal subtype and two belong to the classical subtype. Interestingly, the two cell lines that we predicted to belong to the classical subtype (CAPAN-1 and HPAF-II) have been noted to produce high or moderate amounts of mucin[42,43], which the Moffit et. al paper found to be present in increased levels in the classical subtype[40]. In addition, the Collison et al. publication, whose classical subtype genes significantly overlapped with the Moffit et al. classical subtype genes (20/22), predicted that both CAPAN-1 and HPAF-II belong to the classical subtype[44]. This suggests that these two pancreatic cell lines may indeed reflect the classical subtype, despite the Moffit et al. publication classifying them as basal[40].

Correlations between the pancreatic cell lines and the primary tumors in each individual subtype were also calculated (Supplementary Fig. 4). The rankings of the pancreatic cell lines compared with the primary tumors in the individual subtypes were similar to the rankings of the pancreatic cell lines compared with all of the pancreatic primary tumors, suggesting that global differences between the samples outweigh the subtype-specific differences for PAAD.

**TCGA-110-CL: a comprehensive pan-cancer cell line panel**. The NCI-60 panel of human tumor cell lines has been used in cancer research for almost 30 years to screen chemical compounds and natural products. It contains cell lines from the following ten tumor types: BRCA, COADREAD, GBM, KIRC, LAML, LUAD, LUSC, OV, PRAD, and SKCM. We wanted to determine if the NCI-60 panel could be improved by using cell lines with higher correlations to their primary tumor samples. We analyzed the cell lines that overlapped between the NCI-60 panel and the CCLE database, and found that the cell lines in the NCI-60 panel did not have the highest correlations with their primary tumor samples based on gene expression profiles (Fig. 5a–c). We created an improved NCI-60 panel by selecting the same number of cell lines per tumor type as the original NCI-60 panel, but choosing the cell lines with the highest correlations per tumor type. The correlations in our improved NCI-60 panel were significantly higher than the original NCI-60 panel, which suggests that the integration of primary tumor data can be used to guide cell line selection for more representative models of cancer.

We furthermore propose a new expanded panel of cell lines, which we name TCGA-110-CL, to be used as a pan-cancer resource for cancer research and drug screening (Fig. 5d; Supplementary Data 7). We selected the five cell lines with the highest correlations to their primary tumor samples from each of the 22 tumor types analyzed in this paper to generate our TCGA-110-CL panel. For the nine tumor types for which we have tumor subtype predictions of the cancer cell lines, we select the cell lines with the highest correlation within each tumor subtype to maximize the diversity of tumor subtypes within the panel. By using TCGA primary tumor data to guide our cell line selection, we hope that our new panel will be more comprehensive and representative of primary tumor samples than the NCI-60 panel.

## Discussion
While cell lines are commonly used as models of primary tumors in cancer research, cell lines differ from primary tumors in biologically significant ways and not all cell lines may be appropriate models for their annotated tumor type. Previous studies of ovarian cancer, breast cancer, and liver cancer have shown that the molecular profiles of cell lines from the same tumor type can differ widely and some cell lines more closely model their primary tumors than others. In this study, we leveraged publicly available transcriptomic data to perform a comprehensive pan-cancer analysis across 22 tumor types and provide a resource for researchers to select appropriate cell lines for their tumor-specific studies.

Our analysis reveals that primary tumor and cell line correlations did not vary widely across tumor types. Clustering tumor types by correlations between primary tumor samples and cell lines generally grouped similar tumor types together. Of note, the primary tumor samples in 8/22 tumor types have higher correlation coefficients with cell lines from other tumor types than cell lines from their own tumor type. These tumor types may contain poorly differentiated samples, which would make it difficult to distinguish them from other tumor types using transcriptomics alone.

We identified primary tumor sample purity as a significant confounder in our correlation and differential expression analysis, and show that we are largely able to remove the confounding effect of tumor purity in our analysis. After correcting for primary tumor purity, we found a significantly lower enrichment of immune pathways among the primary tumor samples in our GSEA analysis. We found that cell-cycle-related pathways are consistently upregulated in cell lines across all tumor types, perhaps reflecting in vitro culturing conditions.

In our case study comparing pancreatic cell lines to pancreatic primary tumor samples, we found that the pancreatic cell lines are more representative of pancreatic primary tumor samples than cell lines from other tissues of origin. We also found a group of cell lines with significantly lower correlations with the primary tumors, suggesting that these cell lines may not be appropriate models of primary PAAD tumors. Indeed, the pancreatic cancer cell line with the worst median correlation was shown to express neuroendocrine differentiation[41], and the second lowest cell line was derived from a neuroendocrine tumor rather than an adenocarcinoma. Last, we predicted tumor subtypes for 60% of the pancreatic cell lines, and predicted 15 basal subtype cell lines and 10 classical subtype cell lines to be present in the CCLE. While we presented our analysis of pancreatic cancer here, we also analyzed the other 21 tumor types and present the results in our web application (http://comphealth.ucsf.edu/TCGA110CL).

Finally, we propose the TCGA-110-CL cell line panel as a resource for pan-cancer studies. It encompasses 22 different tumor types and contains the cell lines most correlated with their primary tumor samples. Although some tumor types have higher correlations than others, our aim was to propose a comprehensive cell line panel and we did not set a correlation coefficient cutoff for cell line inclusion. We hope that using more representative cell lines in our pan-cancer panel will improve our ability to translate cell line findings into patients.

There are several limitations of our study that should be recognized. Although we were not able to match all of the cell lines from CCLE to primary tumor samples in TCGA, we were able to match a majority of the cell lines (71%) to a corresponding primary tumor type, and we provide analysis for less common tumor types whose cell lines have not been well studied. In addition, although our cell line findings lack experimental validation, our findings were highly correlated to previous publications[3], and we were able to identify a pancreatic cell line that was derived from a neuroendocrine tumor rather than a PAAD. Last, the focus of our study was on transcriptomics which is only one potential metric for determining cell line suitability, depending on the research question being asked. However, we believe this study is a valuable general resource for researchers who can, for example, use it to identify potentially problematic cell lines that may not be representative of the primary tumors they are studying.

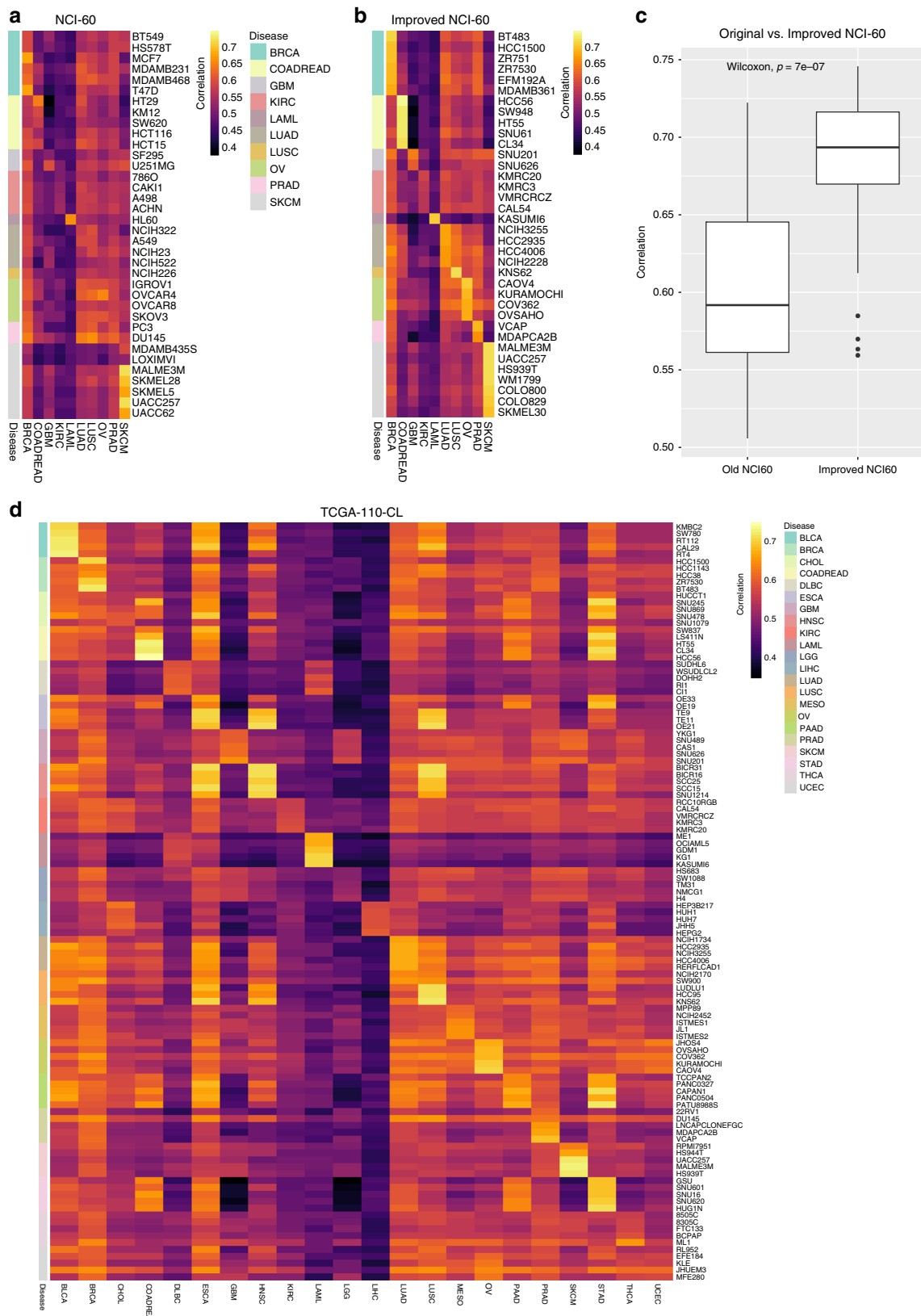

In future studies, we hope to integrate other types of molecular data such as mutation, copy number alteration, and methylation profiles to provide a multi-omic comparison of cell lines and primary tumor samples. In particular, genomic alterations are important for targeted therapies, which act on specific mutant isoforms and we hope to incorporate this information in our future cell line studies.

By leveraging expression profiles from thousands of primary tumor and cell line samples, our study has created a comprehensive pan-cancer resource to aid researchers in selecting the

**Fig. 5** The TCGA-110-CL: an improved cell line panel integrating TCGA and CCLE data. **a** Heatmap of correlations between cell lines in the NCI-60 panel and primary tumor data. Only 36 cell lines which are shared between NCI-60 panel and CCLE are shown. The tumor types of each cell line are indicated by the annotation bar to the left of the heatmap. **b** Heatmap of improved NCI-60 panel. Improved panel has the same number of cell lines and tumor types as the original NCI-60 panel, but the cell lines with the highest correlations with their matched primary tumor samples were selected. **c** Boxplot showing that the improved NCI-60 panel has significantly higher correlations (two-sided Wilcoxon test $p$-value = 7.6e-07) with their matched primary tumor samples compared with the original NCI-60 panel. The center line in the boxplot depicts the median, the box limits depict the upper and lower quartiles, and the whiskers depict 1.5 times the interquartile range (IQR). **d** Proposed TCGA-110-CL panel. An improved cell line panel that includes five cell lines with the highest correlations to their matched primary tumor samples across 22 tumor types. For the tumor types with subtype predictions, cell lines with the highest correlations within each subtype were chosen to maximize subtype representation in the panel

most representative cell line models. We hope that using more appropriate cell line models for cancer studies will allow the research community to better understand cancer biology and translate more in vitro findings into clinically relevant therapies.

## Methods

**Data collection and normalization**. CCLE cell lines were manually matched to TCGA tumor types using the CCLE Cell Line Annotations file (CCLE_sample_info_file_2012–10–18.txt), which contains histological information for each cell line. While 934 CCLE samples were available in the OSF open-access repository, we were able to match ~70% of the samples ($n = 679$) to their respective TCGA tumor type. We used these matched CCLE cell lines for comparison with TCGA primary tumor samples. These samples encompass the following 22 tumor types: BLCA, BRCA, CHOL, COADREAD, DLBC, ESCA, GBM, LGG, HNSC, KIRC, LAML, LIHC, LUAD, LUSC, MESO, OV, PAAD, PRAD, SKCM, STAD, THCA, and UCEC. For the correlation analysis based on cell line tissue of origin, all 934 CCLE samples were used.

TCGA and CCLE RNA-seq samples for the 22 tumor types listed above were downloaded from the Google Cloud Pilot RNA-Sequencing for CCLE and TCGA project in the OSF open-access repository[45] (https://osf.io/gqrz9/). This repository contains 12,307 RNA-seq samples from both the CCLE and the TCGA databases, which have been uniformly processed from raw data. Transcript alignment and quantification were performed using kallisto (version 0.43.0), and both transcript per million (TPM) values and transcript counts are available in the repository. The transcript counts were downloaded and summarized to the gene level for this analysis. We then performed upper-quartile normalization and the log transformed data. Because two different sequencing platforms (GAII and HiSeq) were used by TCGA to sequence five tumor types (UCEC, COADREAD, LAML, STAD, and UCEC), we used ComBat to correct for these sequencing platform differences (Supplementary Fig. 1).

We collected tumor purity estimates for all TCGA samples using the ABSOLUTE[46] method from the TCGA PanCan site (https://gdc.cancer.gov/about-data/publications/pancanatlas). We then computed tumor purity using ESTIMATE[47] for all of the TCGA tumors and averaged the ABSOLUTE and ESTIMATE values. The purity estimates using ABSOLUTE were highly correlated with the purity estimates using ESTIMATE (Supplementary Fig. 2A).

**Correlation analysis**. We analyzed 18,151 protein-coding genes in our correlation analysis. To correct for the heterogeneous cellular composition of the primary tumor samples, we removed genes that have high correlations with tumor purity ($R > −0.4$, adjusted $p$-value $< 0.01$), and adjusted for tumor purity in the primary tumor samples using linear regression. For each tumor type, we then selected the 5000 most variable genes ranked by interquartile range (IQR) across the primary tumor samples only. We decided to use 5000 genes based on previous studies[4], although we tried increasing the number of genes (10000 genes, all genes) and found our results to be remarkably robust (Supplementary Fig. 3A, B). In addition, we performed Gene Ontology analysis on the top 10% (500) of the genes with the highest IQR to understand which biological processes are captured. The results of the GO analysis are presented in Supplementary Data 3.

**Differential expression and GSEA**. We identified differentially expressed genes using limma and voom with quantile normalization. We added tumor purity estimates of the primary tumor samples as covariates, and we set the tumor purity estimates of all the cell lines as 1. We considered a gene to be differentially expressed if the FDR $<0.01$, and the absolute LFC $>2$.

For our GSEA analysis, we ranked our genes by their log fold-change values. We then used the GSEAPreanked[48] software with the classic setting, which was recommended for RNA-seq data in the GSEA manual. The enrichment score (ES) reflects the degree to which a gene set is overrepresented at the top or bottom of the ranked list of genes. We downloaded the 50 Hallmark gene sets from the MSigDB Collections[19], and created our own gmx file for the Hallmarks of cancer pathways using gene sets from the Oncology Models Forum[20].

**Tumor subtype analysis**. We used the Broad Institute's Nearest Template Prediction (NTP)[21] method for our subtype analysis. To generate the subtype templates for each tumor type, we collected subtype information from TCGA publications. We then randomly split the TCGA samples into training (80%) and test set (20%). We used the training set to generate the templates for the NTP method by performing differential expression analysis between each subtype versus all other subtypes with voom and quantile normalization. We selected template genes that had LFC >1 and FDR <0.01 for each subtype. To enrich for cell line relevant genes, we then removed genes that were differentially expressed between cell lines and primary tumors with LFC >2 and genes that were not in the top 50% of expression in at least two cell lines. Next, we used these filtered subtype templates to predict the subtypes of the primary tumors held out in the test set using the NTP method. If the classification accuracy in the test set was ≥80%, we then applied it to the cell lines to predict the cell line subtypes.

**Reporting summary**. Further information on research design is available in the Nature Research Reporting Summary linked to this article.

## Data availability

All data used in this study are publicly available. The TCGA and CCLE RNA-seq count matrixes were originally downloaded from the Google Cloud Pilot RNA-sequencing for CCLE and TCGA open-access repository: https://osf.io/gqrz9. The normalized expression data used in this study is available on SynapseSynapse (https://www.synapse.org) under Synapse ID syn18685536. Tumor purity estimates for all TCGA samples using the ABSOLUTE method were downloaded from the TCGA PanCanAtlas publications website: https://gdc.cancer.gov/about-data/publications/pancanatlas. GSEA hallmark gene sets were downloaded from the GSEA MSigDB Collections website: http://software.broadinstitute.org/gsea/msigdb/collections.jsp. The hallmarks of cancer gene sets were downloaded from the Oncology Model Fidelity Score GitHub page: https://github.com/tedgoldstein/hallmarks.

## Code availability

The code for normalization and comparing the TCGA and CCLE gene expression profiles is available at https://github.com/katharineyu/TCGA_CCLE_paper.

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

## Acknowledgements

Butte, Aran, and Goldstein graciously acknowledge the support from the National Cancer Institute Oncology Model Forum project, NIH grant U24 CA195858. D.A. is supported by the Gruss Lipper Postdoctoral Fellowship, B.C. by R21 TR001743 and K01 ES028047, and M.S. by National Library of Medicine NIH grant K01 LM012381. We would like to thank Boris Oskotsky for technical support and members of the Sirota Lab for useful discussion. We would also like to thank researchers at the Broad Institute and The Cancer Genome Atlas Consortium who released data to the public. The content is solely the responsibility of the authors, and does not necessarily represent the official views of the National Institutes of Health.

## Author contributions

Conceptualization, A.J.B., B.C., D.A., and M.S.; investigation, K.Y.; software, K.Y., B.C., and J.C.; writing of the original draft, K.Y. and M.S.; discussion. Writing—review and editing, K.Y., B.C., D.A., J.C., C.Y., D.W., L.J.V., A.J.B., T.G., and M.S.; supervision, M.S. and T.G.

## Additional information

**Competing interests:** The authors declare no competing interests.

