## [Peer Review File · Nature Communications]

Reviewers' comments:

Reviewer #1 (Remarks to the Author):

In this manuscript Yu and colleagues used correlation of expression data between TCGA tumor types and matched tumor types from the CCLE to identify the best representative cell line to model cancer types. This is an important question as cell lines do not always recapitulate the primary cancer based on culturing methods or selective pressures of culture. I think this paper has value, but I do have some concerns with basing on total correlation to the TCGA tumors.

My major concern is that there is too much emphasis on maximum correlation between a cell line and all of the tumors. While the manuscript mentions that they looked at subtypes within tumor subtypes, the overall assumption for defining the TCGA-110 was based on max correlation. This is concerning when there is tumor heterogeneity and potentially bias in the types of tumors that might have been profiled. In the COAD/READ example that was highlighted, while the text commented on highest correlation with CMS1, three of the selected lines in the TCGA-110 were more correlated with CMS2. And there was a highly correlated samples with the CMS4, but it wasn't selected based on the requirement for high correlation. Another example is the BRCA cancers. ER-negative cancers are very distinct molecularly from ER-positive breast cancers, but as 80% of TCGA BRCA was ER-positive, they would dominate the correlations. While one ER-negative tumor make the final 5, it wasn't even correlated with basal-like tumors in your online supplement, and doesn't fully capture the diversity within the TNBC. All five of the final BRCA samples most correlated with luminal subtypes. The paper mentions "This suggests the importance of considering tumor subtype information when selecting cell lines for cancer studies." However, it seems as if that wasn't used for the selection of the TCGA-110.

I also am curious why you allowed 5 cell lines per tumor type, even when there was less correlation in a given tumor type. The analysis with the NCI-60 and improved NCI-60 showed that the correlations overall improved from 0.6 to 0.7 medians. However, for several of the tumor types like UCEC, THCA, LICH, CHOL were mostly less than 0.6 correlations. Would it have been better to just comment that there aren't great cell lines models than to have some tumor types with cell lines listed that are suboptimal?

Which 6 tumor types had higher correlations with tumor cell lines of other origins? It was difficult to tell with the resolution of figure 1C.

For figure 3, I thought it was great that your analysis was able to pick out cell lines that were determined to be derived from fibroblasts instead of the colon or rectum adenocarcinoma. However in the part where you integrate into the COADREAD subtypes, it comes back to my major issue that your analysis is almost forcing the cell lines to all correlate to all samples. Figure 3C to me just shows that some cell lines are more correlated with each other than all tumors vs showing a correlation with the subtypes. The same with the boxplot in figure 3D. What I did find interesting and maybe should be more of a focus was the heatmap in figure 3D. As you mentioned, many of the cell lines are more correlated with CMS1, BUT there are still 6-7 cell lines that are highly correlated with CMS2 and very anti correlated with CMS1. There is also one cell line, HS255T, that is highly correlated with CMS4 and no cell lines that best model the CMS3. Though interestingly, in your final TCGA-110, you selected mostly CMS2 enriched samples but would not include a model of CMS4 since it wasn't correlated with most of the other tumor.

Did the cell lines that were identified here, match with the prior studies mentioned in the introduction that also addressed these types of analyses in individual tumor type settings.

In your batch correction, did you also take into account the Illumina Sequencer information? Particularly in the COAD/READ samples from TCGA as well as UCEC, AML, and STAD, a subset of the samples were performed on Illumina GAII sequencers and others on Illumina HiSeq. In the recent TCGA pancancer papers, they noted there were batch issues between the sequencers. It would be good to make sure this wasn't an issue as well.

For the purity analysis, the pathology metrics of purity rarely correlate with the molecular purity. For your purity estimates, you took the median value of the purity from 2 or more methods. Did you look to see the comparison of the purity measurements? Also for the correlation by purity, you used the top quartile vs bottom quartile. However, the purity for some tumor types varied widely so that the quartiles from one tumor type would not be relatable to another tumor type. For example, the PAAD had significantly lower purity than other samples in TCGA. That information was only found in a supplemental figure legend. It would be good to add to the methods.

Supplemental Table 1 – did you only include the primary tumors (or primary/mets for SKCM)? The TCGA breast cancer number seems high and possibly includes the adjacent normal? It would be beneficial to include a supplemental table with the IDs used for both the TCGA tumor types and the CCLE cell lines.

Reviewer #2 (Remarks to the Author):

In this manuscript, Yu et al. present a systematic comparison of available gene expression profiles of cancer cell lines (from the Cancer Cell Line Encyclopedia) and primary tumors (from The Cancer Genome Atlas) in order to identify the cell lines that closely represent the primary tumors and, therefore, are more suitable models. Evaluating the suitability of cell line models of cancer is an important question with great implications in the field. However, I believe this paper falls short of providing a convincing methodology that can be widely adopted. Specifically, I have deep reservations regarding the use of gene expression similarity as the sole metric for ranking cell line suitability, and I believe more analyses and/or experiments are required before the reliability of this metric can be assessed. In other words, what needs to be first established is the assumption that underlies this study: that gene expression similarity can be used as a metric for identifying suitable cell lines. I also have additional conceptual and technical concerns, as outlined below.

Major conceptual concerns:

1. The authors use gene expression similarity as a metric to rank the available cell line models of each cancer type. However, no systematic evidence is provided that this ranking can separate cell lines that are appropriate models of each cancer from cell lines that are less suitable. In fact, some of the presented data suggests otherwise – for example, on page 4 the authors say “interestingly, we observed that sometimes the highest correlation coefficients are not necessarily between cell line and primary tumor samples from the same tumor type”. If gene expression similarity is not even able to assign the cell lines to their respective cancer type, one might suspect that it may also not be able to rank the cell lines within each cancer type. If the authors could devise a strategy to actually show, in a systematic way, that their transcriptomics-based metric identifies better cell line models, then this study would be seen in a completely different light.

2. The factors that determine the suitability of a cell line depend on the biological question that is being asked, and I suspect it would be difficult to define a one-size-fits-all metric of suitability. For example, some drugs target specific mutant isoforms of proteins (for instance BRAF V600E inhibitors), and therefore presence of these mutations would be far more important than gene expression when assessing the functions of these drugs. In my opinion, gene expression would become important only secondary to these other factors; for example, if there are multiple cell lines that harbor the suitable BRAF mutations, then it might be the case that among those options the one that better reflects the primary tumor expression is more suitable.

3. It is not clear to me to what extent the gene expression similarities are driven by modules that are important for the specific biology of each cancer type. For example, the authors mention that cell cycle genes are often up-regulated in cell lines compared to primary tumors. Therefore, one might expect that the presented method would in fact select for cell lines that have similar cell cycle expression to the primary tumor, at the cost of ignoring other pathways that are important for each cancer type. Can the authors show which genes are driving the correlations between cell lines and primary tumor samples?

4. Are the cell lines that better match the primary tumors at the transcriptomic level also more closely related to the primary tumors from the perspective of genomic alterations? Some of the cell lines in the authors' TCGA-110 panel have been previously shown to contain none of the mutations that are hallmarks of their respective cancers. I believe a systematic examination of genomic alterations in at least one of the cancer types is warranted.

Major technical concerns:

1. Central to the methodology that is used by the authors is normalization using RUVg (Remove Unwanted Variation Using Control Genes). RUVg requires a set of control genes, which the authors determine empirically. However, the details are not provided. Specifically, the RUVg method requires knowing the sources of "wanted variation" (e.g. two biological conditions), identifying genes that are not associated with these sources of wanted variation (control genes), and then using these genes to identify the sources of "unwanted variation". It is however difficult to understand how this approach fits into the study design presented in this paper. Did the authors use genes that are not differentially expressed between primary tumors and cell lines? In that case, did they use different sets of control genes for each cancer type? Since the batch information for TCGA data is available (<https://bioinformatics.mdanderson.org/main/TCGABatchEffects:Overview>), can the authors show that the sources of unwanted variation that RUVg identifies actually correlate with the TCGA batches?

2. Another central aspect of the methodology is correcting for tumor purity, which the authors estimate as the median of values reported by four different methods. Not all samples had estimates across this ensemble – therefore, using the median of the available estimates can potentially introduce further batch effects via purity adjustment. Would the authors kindly address this concern?

3. It is not clear whether the authors have used log-transformed values in different steps of their analysis or the original normalized gene expression measurements. For example, using untransformed data in the linear regression for purity correction would mean that large outliers could dominate the regression. Overall, technical details such as this should be indicated explicitly in the Methods section.

Other comments:

1. Would the authors kindly clarify whether the values in supplementary figure 1c were computed before or after RUVg normalization? Could they show the results for both cases?
2. Did the authors account for the mean-variance relationship in RNA-seq gene expression when selecting the top 5000 features for their correlation analyses? Also, the authors indicate that their study was robust to the choice of the number of genes included in the correlation. However, the figure that they cite (supplementary figure 4) simply shows the distribution of correlations, but not the actual underlying correlations. Would the TCGA-110 panel change if different genes were used for the correlation analysis?
3. Why use the geometric mean in figure 1c?
4. I believe the CCLE contains cell lines that were derived from each other and are hence highly similar. Did the authors notice this and did they include these potentially redundant cell lines in their TCGA or NCI panels?
5. A portion of the legend labels for supplementary figure 1b is hidden.
6. I am generally concerned about the reproducibility of this study – many of the critical details in the methods are missing, and no code or data are provided to reproduce the results.
7. In addition to the underlying codes and datasets that were used to carry out this study, some of the most important outputs of this study are not provided. A critical example is the matrix of cell line/primary tumor correlations. The authors should provide a complete matrix of 9111x666 pairwise correlations between the TCGA primary tumors and the CCLE cell lines.

Reviewers' comments:

Reviewer #1 (Remarks to the Author):

In this manuscript Yu and colleagues used correlation of expression data between TCGA tumor types and matched tumor types from the CCLL to identify the best representative cell line to model cancer types. This is an important question as cell lines do not always recapitulate the primary cancer based on culturing methods or selective pressures of culture. I think this paper has value, but I do have some concerns with basing on total correlation to the TCGA tumors.

We would like to thank the reviewer for a nice summary and for recognizing the value of the presented work. We address the specific concerns below.

My major concern is that there is too much emphasis on maximum correlation between a cell line and all of the tumors. While the manuscript mentions that they looked at subtypes within tumor subtypes, the overall assumption for defining the TCGA-110 was based on max correlation. This is concerning when there is tumor heterogeneity and potentially bias in the types of tumors that might have been profiled. In the COAD/READ example that was highlighted, while the text commented on highest correlation with CMS1, three of the selected lines in the TCGA-110 were more correlated with CMS2. And there was a highly correlated sample with the CMS4, but it wasn't selected based on the requirement for high correlation. Another example is the BRCA cancers. ER-negative cancers are very distinct molecularly from ER-positive breast cancers, but as 80% of TCGA BRCA was ER-positive, they would dominate the correlations. While one ER-negative tumor made the final 5, it wasn't even correlated with basal-like tumors in your online supplement, and doesn't fully capture the diversity within the TNBC. All five of the final BRCA samples most correlated with luminal subtypes. The paper mentions "This suggests the importance of considering tumor subtype information when selecting cell lines for cancer studies." However, it seems as if that wasn't used for the selection of the TCGA-110.

We thank the reviewer for their comments and have expanded the tumor subtype analysis to predict tumor subtypes based on the Nearest Template Prediction (NTP) method from the Broad Institute, which we believe is better able to distinguish between subtypes than our original highest correlation approach. The text describing our new subtype analysis method (page 6) is provided below and in Figure 3A:

"In order to predict the subtype of individual cancer cell lines, we used the Broad Institute's Nearest Template Prediction (NTP) method which has previously been used to predict the subtypes of cancer cell lines. Briefly, this method involves generating gene templates for each subtype by identifying genes that are upregulated in each subtype compared to the other subtypes. The distances between the sample to be classified and each subtype template is then calculated and the sample is predicted to belong to the subtype with the smallest template distance.

Like Sveen et. al, we modified this method to create a classifier that can be applied to cancer cell lines after training the classifier on primary tumor samples. We began with the 18 TCGA tumor types for which we had subtype information from TCGA publications and randomly divided these samples into training sets (80%) and test sets (20%). After generating our initial subtype templates using the training set of primary tumor samples, we removed genes that are differentially expressed between primary tumors and cell lines as we wanted to enrich our subtype templates for genes that are consistent between primary tumors and cell line models. We also filtered out genes that are not highly expressed in at least a subset of the cell lines as we wanted to retain genes that are robust and informative in cell lines. This filtering step can also enrich for cancer-intrinsic genes since cell lines are pure populations of cancer cells. To verify that our classifier is still able to predict tumor subtypes after enriching for cell line-relevant genes, we applied the classifier to the test set of held out primary tumor samples. 9/18 tumor types had a classification accuracy greater than or equal to 80% in the test set. We then applied the classifiers of these 9 tumor types to their respective cell lines and predicted the subtypes of the individual cell lines.”

For the new TCGA-110 cell line panel, we have selected cell lines with the highest correlations within each tumor subtype to capture the diversity of the tumors. The new TCGA-110 panel (which includes tumor subtype information when appropriate) is provided in Supplementary Table 7. We also added the following text to the results section describing the TCGA-110 panel on page 9:

“For the 9 tumor types for which we have tumor subtype predictions of the cancer cell lines, we select the cell lines with the highest correlation within each tumor subtype to maximize the diversity of tumor subtypes within the panel.”

I also am curious why you allowed 5 cell lines per tumor type, even when there was less correlation in a given tumor type. The analysis with the NCI-60 and improved NCI-60 showed that the correlations overall improved from 0.6 to 0.7 medians. However, for several of the tumor types like UCEC, THCA, LICH, CHOL were mostly less than 0.6 correlations. Would it have been better to just comment that there aren't great cell lines models than to have some tumor types with cell lines listed that are suboptimal?

We thank the reviewer for their comments. Rather than set an arbitrary cutoff value for 'good' cell lines, we chose the top 5 cell lines per tumor type because our aim was to propose a comprehensive cell line panel covering a broad range of tumor types, however all the data is available for researchers to choose cell lines and explore the results as part of the online tool that we provide. We have added a sentence to the discussion stating that some tumor types have better models than others in our proposed panel. The added text (on page 10) is provided below:

“Although some tumor types have higher correlations than others, our aim was to propose a comprehensive cell line panel and we did not set a correlation coefficient cutoff for cell line inclusion.”

Which 6 tumor types had higher correlations with tumor cell lines of other origins? It was difficult to tell with the resolution of figure 1C.

We apologize for the low resolution of figure 1C and have adjusted the font/resolution to make sure it is more readable. In the new analysis, 8 tumor types had correlations with tumor cell lines of other origins and are listed here: ESCA (HNSC), BLCA (HNSC), STAD (COADREAD), LUSC (HNSC), CHOL (LIHC), LUAD (PAAD), UCEC (OV), LGG (GBM)

Many of the tumor types listed above have higher correlations with highly related tumor types (e.g. ESCA and HNSC, STAD and COADREAD, LGG and GBM, UCEC and OV, STAD and COADREAD, etc.)

For figure 3, I thought it was great that your analysis was able to pick out cell lines that were determined to be derived from fibroblasts instead of the colon or rectum adenocarcinoma. However in the part where you integrate into the COADREAD subtypes, it comes back to my major issue that your analysis is almost forcing the cell lines to all correlate to all samples. Figure 3C to me just shows that some cell lines are more correlated with each other than all tumors vs showing a correlation with the subtypes. The Same with the boxplot in figure 3D. What I did find interesting and maybe should be more of a focus was the heatmap in figure 3D. As you mentioned, many of the cell lines are more correlated with CMS1, BUT there are still 6-7 cell lines that are highly correlated with CMS2 and very anti correlated with CMS1. There is also one cell line, HS255T, that is highly correlated with CMS4 and no cell lines that best model the CMS3. Though interestingly, in your final TCGA-110, you selected mostly CMS2 enriched samples but would not include a model of CMS4 since it wasn't correlated with most of the other tumor.

In the new analysis, we have taken tumor subtype into account when selecting cell lines for the TCGA-110 panel as well as when describing our case study. We have done our best to create a panel that captures the diversity of the tumor types by maximizing the tumor subtype coverage.

In the revised version, we have modified the case study to highlight pancreatic adenocarcinoma (PAAD) because we found a recent paper from Sveen et al. (Clin Can Res 2018) characterizing the COADREAD cell lines.

Did the cell lines that were identified here, match with the prior studies mentioned in the introduction that also addressed these types of analyses in individual tumor type settings.

Yes, the relative order of the cell lines in our study were largely consistent with the previous publications cited in the introduction, even among papers that used other omics data types

such as copy number and mutation data. We have added a few sentences about this in the results on page 5. The new text is also provided here:

“To verify that the results of our transcriptomic-based correlation approach were consistent with previous publications, we compared our cell line rankings for OV to the cell line rankings in Domcke et al. which evaluated high grade ovarian cancer cell lines based on copy number alterations and selected mutations. Our results were highly correlated (Spearman’s rho = 0.6, p-value = 5.837e-05), which suggests that our cell line rankings capture much of the same information as more curated ranking methods that use genomic alterations.”

In your batch correction, did you also take into account the Illumina Sequencer information? Particularly in the COAD/READ samples from TCGA as well as UCEC, AML, and STAD, a subset of the samples were performed on Illumina GAII sequencers and others on Illumina HiSeq. In the recent TCGA pancancer papers, they noted there were batch issues between the sequencers. It would be good to make sure this wasn’t an issue as well.

We thank the reviewer for this comment. We did not take into account the Illumina Sequencer information in our initial analysis but have since corrected for sequencing platform-based effects within TCGA using ComBat. This is now described on page 10 of the methods section and in Supplementary Figure 1. The new text is provided below:

“Because two different sequencing platforms (GAII and HiSeq) were used by TCGA to sequence 5 tumor types (UCEC, COADREAD, LAML, STAD, UCEC), we used ComBat to correct for these sequencing platform differences”

For the purity analysis, the pathology metrics of purity rarely correlate with the molecular purity. For your purity estimates, you took the median value of the purity from 2 or more methods. Did you look to see the comparison of the purity measurements? Also for the correlation by purity, you used the top quartile vs bottom quartile. However, the purity for some tumor types varied widely so that the quartiles from one tumor type would not be relatable to another tumor type. For example, the PAAD had significantly lower purity than other samples in TCGA. That information was only found in a supplemental figure legend. It would be good to add to the methods.

We thank the reviewer for their comments. In our re-analysis, we have selected two methods for estimating tumor purity (ABSOLUTE and ESTIMATE) because the results of these two methods are relatively well correlated (R = 0.6, Supplementary fig. 2A) and we have complete purity information for all our TCGA samples with these two methods, which reduces the bias caused by incomplete tumor purity information. The new text describing our purity estimates is provided below:

“We collected tumor purity estimates for all TCGA samples using the ABSOLUTEⁱ method from the TCGA PanCan site (<https://gdc.cancer.gov/about-data/publications/pancanatlas>). We then computed tumor purity using ESTIMATEⁱⁱ for all of the TCGA tumors and averaged the ABSOLUTE

and ESTIMATE values. The purity estimates using ABSOLUTE were highly correlated with the purity estimates using ESTIMATE (Supplementary Figure 2A).”

We apologize for the confusion with the correlation by purity figure. We compared the top quartile vs bottom quartile purity measurements within each tumor type separately so the quartile measurements of one tumor type do not affect the other tumor types. We have expanded upon the purity analysis in the methods section on page 10 and have added a figure summarizing the purity estimates across the tumor types (Supplementary Figure 2B).

Supplemental Table 1 – did you only include the primary tumors (or primary/mets for SKCM)? The TCGA breast cancer number seems high and possibly includes the adjacent normal? It would be beneficial to include a supplemental table with the IDs used for both the TCGA tumor types and the CCLE cell lines.

We apologize for the oversight. Primary tumors were used in all tumor types except for SKCM, in which case the metastatic tumors were included because the SKCM TCGA cohort was primarily focused on metastatic tumors. While only the tumor samples were used in the analysis, supplementary Table 1 included both the tumors and adjacent normal samples. We have corrected the table to only include the tumor samples. The TCGA IDs and CCLE cell lines used has been added as a supplemental Table 3.

Reviewer #2 (Remarks to the Author):

In this manuscript, Yu et al. present a systematic comparison of available gene expression profiles of cancer cell lines (from the Cancer Cell Line Encyclopedia) and primary tumors (from The Cancer Genome Atlas) in order to identify the cell lines that closely represent the primary tumors and, therefore, are more suitable models. Evaluating the suitability of cell line models of cancer is an important question with great implications in the field. However, I believe this paper falls short of providing a convincing methodology that can be widely adopted. Specifically, I have deep reservations regarding the use of gene expression similarity as the sole metric for ranking cell line suitability, and I believe more analyses and/or experiments are required before the reliability of this metric can be assessed. In other words, what needs to be first established is the assumption that underlies this study: that gene expression similarity can be used as a metric for identifying suitable cell lines. I also have additional conceptual and technical concerns, as outlined below.

We would like to thank the reviewer for recognizing the value of the question that we chose to address. We agree that gene expression is just one metric of identifying suitable cell lines. However, we believe gene expression is especially valuable for evaluating cell line suitability because it is downstream of genetic alterations such as mutations or copy number changes. Indeed, previous publications have shown that the functional consequences of genetic alterations are reflected in the expression profiles. For example, Kim et al. (Cell Systems 2017) isolated a BRAF V600E transcriptional activation signature and decomposed the KRAS G12V activation signature into multiple transcriptional components that reflect the activities

downstream of RAS. Additionally, Miller et al. (PNAS 2005) generated a p53 mutant gene expression signature in breast cancer that was shown to be superior to p53 mutation status alone in predicting prognosis. Other publications have shown that mutation status is insufficient in accurately predicting drug response to targeted agents and that transcriptional activity could often explain the variability in response. Konieczkowski et al. (Cancer Discovery 2014) found that, among melanoma patients who had the “actionable” BRAF V600 mutation, those who failed to respond to MAPK inhibitors had distinct transcriptional profiles that were different from those who were drug-sensitive. Similarly, Singh et al. (Cancer Cell 2009) show that cancers with KRAS mutations can be divided into KRAS-dependent and KRAS-independent groups based on their transcriptional profiles. As gene expression captures the functional consequences of upstream mutations and provides a more global snapshot of the cellular state, we believe it is reasonable to evaluate the cell lines based on their expression profiles.

Additionally, we have shown in our colorectal and pancreatic case studies that we are able to identify cell lines that are derived from different tumor types than the ones profiled by TCGA, which suggests that our transcriptomic approach is able to identify potentially problematic cell lines. Per suggestion of the first reviewer, we have also decided to include additional analysis on tumor subtypes which are commonly defined based on expression^{1 2 3}. We have added some text to the discussion describing the shortcomings of using gene expression alone, as this is an important point made by the reviewer. However, for the purposes of this work we choose to focus on gene expression based on the reasons described above.

¹ Mischel, P. S. et al. Identification of molecular subtypes of glioblastoma by gene expression profiling. *Oncogene* 22, 2361–2373 (2003).

² Tothill, R. W. et al. Novel Molecular Subtypes of Serous and Endometrioid Ovarian Cancer Linked to Clinical Outcome. *Clinical Cancer Research* 14, 5198–5208 (2008).

³ Sorlie, T. et al. Gene expression patterns of breast carcinomas distinguish tumor subclasses with clinical implications. *Proceedings of the National Academy of Sciences* 98, 10869–10874 (2001).

Major conceptual concerns:

1. The authors use gene expression similarity as a metric to rank the available cell line models of each cancer type. However, no systematic evidence is provided that this ranking can separate cell lines that are appropriate models of each cancer from cell lines that are less suitable. In fact, some of the presented data suggests otherwise – for example, on page 4 the authors say “interestingly, we observed that sometimes the highest correlation coefficients are not necessarily between cell line and primary tumor samples from the same tumor type”. If gene expression similarity is not even able to assign the cell lines to their respective cancer type, one might suspect that it may also not be able to rank the cell lines within each cancer type. If the authors could devise a strategy to actually show, in a systematic way, that their

transcriptomics-based metric identifies better cell line models, then this study would be seen in a completely different light.

We thank the reviewer for the important point that is brought up here. In the original COADREAD case study, we were able to identify cell lines that were derived from fibroblasts rather than epithelial tumor cells due to their lower correlations with the primary tumors. In the revised version, we have modified the case study to highlight pancreatic adenocarcinoma (PAAD) and were also able to identify a cell line that was derived from a neuroendocrine tumor rather than a pancreatic adenocarcinoma. We believe these results support the value of using gene expression similarity as a metric for identifying suitable cell lines. Additionally, our ovarian cell lines ranked by median correlation coefficients are significantly correlated (Spearman's $\rho = 0.6$, p -value = $5.837e-05$) to a previously published ranked list of ovarian cell lines where the suitability scores were derived from copy number alterations and curated mutation patterns (<https://www.nature.com/articles/ncomms3126>). This has been added to the manuscript on page 8. This suggests that our cell line rankings capture much of the same information as more curated ranking methods based on other data types such as mutation and copy number data. Lastly, gene expression contains important information about pathway activity and downstream effects of genomic alterations and is commonly used to stratify tumors into clinically relevant subtypes, which we believe justifies our use of expression data to rank cell lines.

2. The factors that determine the suitability of a cell line depend on the biological question that is being asked, and I suspect it would be difficult to define a one-size-fits-all metric of suitability. For example, some drugs target specific mutant isoforms of proteins (for instance BRAF V600E inhibitors), and therefore presence of these mutations would be far more important than gene expression when assessing the functions of these drugs. In my opinion, gene expression would become important only secondary to these other factors; for example, if there are multiple cell lines that harbor the suitable BRAF mutations, then it might be the case that among those options the one that better reflects the primary tumor expression is more suitable.

Thank you for bringing up this point and example. We completely agree that there are many factors that determine the suitability of a cell line and that very much depends on the question that is being asked. We have added this point to the discussion on pages 9-10. Our aim is not to create a one-size-fits-all metric, but to provide a general resource for researchers who can, for example, use it to identify potentially problematic cell lines that may not be representative of the primary tumors they are studying. While mutation data is certainly important for targeted drugs, we believe it is outside the scope of this study but will be a very useful extension in the future. For researchers who are interested in studying specific mutant isoform inhibitors (such as BRAF V600E inhibitors, as the reviewer suggests), we recommend that they identify cell lines with the relevant mutations of interest, perhaps using a resource such as the Wellcome Sanger Institute's Cell Lines Project browser, and then perhaps use our resource to determine which of these cell lines are most suitable for each tumor type.

3. It is not clear to me to what extent the gene expression similarities are driven by modules

that are important for the specific biology of each cancer type. For example, the authors mention that cell cycle genes are often up-regulated in cell lines compared to primary tumors. Therefore, one might expect that the presented method would in fact select for cell lines that have similar cell cycle expression to the primary tumor, at the cost of ignoring other pathways that are important for each cancer type. Can the authors show which genes are driving the correlations between cell lines and primary tumor samples?

We apologize for the confusion in our methods - we used the top 5,000 most variable genes for each tumor type but only among the primary tumor samples (rather than both primary tumors and cell lines) so we expect that the correlations will be driven by the genes that are important to the specific biology of each tumor type and not cell line/primary tumor differences.

We have added Supplementary Table 7 which presents the top 10% (500) genes from each tumor type that are driving the correlations. We identify these genes by sorting the 5000 most variable genes from each tumor type by the median difference in ranks squared (d^2) from the cell line/primary tumor comparisons. In other words, we interpret the genes with the highest d^2 values as the ones driving the correlations. For each tumor type, we then performed GO analysis on these genes to understand which biological processes are driving the correlations. These are also presented in Supplementary Table 7. In general, many developmental pathways were enriched, which makes biological sense since cancer often involves the reactivation/alteration of developmental processes. This has been added to the manuscript on page 11.

4. Are the cell lines that better match the primary tumors at the transcriptomic level also more closely related to the primary tumors from the perspective of genomic alterations? Some of the cell lines in the authors' TCGA-110 panel have been previously shown to contain none of the mutations that are hallmarks of their respective cancers. I believe a systematic examination of genomic alterations in at least one of the cancer types is warranted.

This is a very interesting question, however we believe that this is outside the scope of the current study in which our focus is on gene expression with the new focus on tumor subtypes as suggested by reviewer 1. We have added a sentence in the discussion on the importance of genomic alterations to match cell lines in future studies on page 10.

Major technical concerns:

1. Central to the methodology that is used by the authors is normalization using RUVg (Remove Unwanted Variation Using Control Genes). RUVg requires a set of control genes, which the authors determine empirically. However, the details are not provided. Specifically, the RUVg method requires knowing the sources of "wanted variation" (e.g. two biological conditions), identifying genes that are not associated with these sources of wanted variation (control genes), and then using these genes to identify the sources of "unwanted variation". It is however difficult to understand how this approach fits into the study design presented in this paper. Did the authors use genes that are not differentially expressed between primary tumors

and cell lines? In that case, did they use different sets of control genes for each cancer type? Since the batch information for TCGA data is available (<https://bioinformatics.mdanderson.org/main/TCGABatchEffects:Overview>), can the authors show that the sources of unwanted variation that RUVg identifies actually correlate with the TCGA batches?

We thank the reviewer for pointing out the presence of batch effects within TCGA, which was not corrected for with RUVg. We have corrected for the TCGA batch effects based on different sequencing platforms using ComBat in a method similar to the TCGA consortia in their pan-cancer paper ([https://www.cell.com/cell/fulltext/S0092-8674\(18\)30302-7#secsectitle0075](https://www.cell.com/cell/fulltext/S0092-8674(18)30302-7#secsectitle0075)) and have decided not to use RUVg to remove cross-study batch effects as concerns were raised about excessively removing biological variability.

2. Another central aspect of the methodology is correcting for tumor purity, which the authors estimate as the median of values reported by four different methods. Not all samples had estimates across this ensemble – therefore, using the median of the available estimates can potentially introduce further batch effects via purity adjustment. Would the authors kindly address this concern?

We thank the reviewer for their comments. To avoid additional batch effects, we have since focused our analysis on purity estimates derived using the ABSOLUTE and ESTIMATE methods since we have purity estimates from both of these methods across 20 tumor types. We have excluded DLBC and LAML from the purity analysis because we are unable to estimate tumor purity for these tumor types using ESTIMATE which is based on gene expression of immune genes.

3. It is not clear whether the authors have used log-transformed values in different steps of their analysis or the original normalized gene expression measurements. For example, using untransformed data in the linear regression for purity correction would mean that large outliers could dominate the regression. Overall, technical details such as this should be indicated explicitly in the Methods section.

We apologize for this oversight. The gene expression measurements have been log transformed and upper-quartile normalized prior to performing linear regression for tumor purity correction. This and additional details have been added to the Methods section on page 10.

Other comments:

1. Would the authors kindly clarify whether the values in supplementary figure 1c were computed before or after RUVg normalization? Could they show the results for both cases?

The figures in S1C were computed before RUVg normalization. We are happy to provide the results for both cases, although we have decided to not use RUVg in the resubmitted version

due to the concerns that were brought up by the reviewer and it is no longer relevant to our analysis.

2. Did the authors account for the mean-variance relationship in RNA-seq gene expression when selecting the top 5000 features for their correlation analyses? Also, the authors indicate that their study was robust to the choice of the number of genes included in the correlation. However, the figure that they cite (supplementary figure 4) simply shows the distribution of correlations, but not the actual underlying correlations. Would the TCGA-110 panel change if different genes were used for the correlation analysis?

While we did not adjust for the mean-variance relationship when selecting the top 5000 features for the correlation analysis, increasing the number of genes to 10,000 genes and all genes does not greatly affect the ranks of the cell lines in each tumor type. We have added additional plots showing the underlying correlation coefficients using 5000 IQR genes compared to 10000 IQR genes and 5000 IQR genes compared to all genes as supplementary fig 3A-B.

3. Why use the geometric mean in figure 1c?

We apologize for this oversight – we have changed Figure 1C to use median correlation values.

4. I believe the CCLE contains cell lines that were derived from each other and are hence highly similar. Did the authors notice this and did they include these potentially redundant cell lines in their TCGA or NCI panels?

Thank you for bringing up this important point - identical cell lines were removed to the best of our ability prior to analysis.

5. A portion of the legend labels for supplementary figure 1b is hidden.

Thank you, this has been corrected.

6. I am generally concerned about the reproducibility of this study – many of the critical details in the methods are missing, and no code or data are provided to reproduce the results.

We apologize for the lack of details and have updated the methods section to include more details about the analysis. We are happy to provide the code and the data – all the results are available as a web tool.

7. In addition to the underlying codes and datasets that were used to carry out this study, some of the most important outputs of this study are not provided. A critical example is the matrix of cell line/primary tumor correlations. The authors should provide a complete matrix of 9111x666 pairwise correlations between the TCGA primary tumors and the CCLE cell lines.

We thank the reviewer for pointing out this oversight and have since added a complete matrix of all pairwise correlations in the supplemental data.

Reviewers' comments:

Reviewer #1 (Remarks to the Author):

I think the addition of the subtype information is very useful. For the subtype comparison versus your all tumor comparison, did you see major differences in correlation by adding in the tumor subtype information?

The purity correction using an updated sets of purity seems to have improved the analysis.

Supp table 4 shouldn't it also have a ranking measurement more than just an FDR score

Supplementary table 5 show the individual pairwise correlations, but it seems like it also needs the correlation to the tumor type or tumor subtype as a whole to be easily usable by other users.

Having looked at the supplement table 6 (top 500 genes and GO pathways, different from the table number listed in the response to reviewers), I'm now a little concerned about the genes used for correlation and the 500 genes selected for each tumor type. For example, in BRCA the top Biological Processes are extracellular matrix related which is not tumor-related and in the 500 genes, I'm not seeing standard well known markers like ESR1, ERBB2, etc.

Reviewer #2 (Remarks to the Author):

I thank the authors for their revised manuscript. My major concerns have been addressed – most importantly, the new comparison of OV cell lines to a previous ranking based on an orthogonal approach provides evidence that the authors expression-based approach is in fact able to pick cell lines that reflect the characteristics of their intended tumours. Also, the suggestion by reviewer #1 to perform separate analysis of subtypes was excellent, and it appears to me that this other major addition has been done properly by the authors (method is appropriate and cross-validation seems to support the utility of the approach). Other technical concerns such as batch effect correction also seem to have been addressed. I have a few minor comments that I hope would further improve the manuscript:

1. As I mentioned, the authors have now included a comparison of their rankings for OV cell lines to a previous publication, but have only mentioned the Spearman correlation and its p-value. Can the authors kindly provide a supplementary table with the information for these cell lines, their rankings in the previous publication that was cited, and their expression correlation based on the author's approach?

2. Figure 5: The panel labels need to be corrected (panels should be A-D).

3. Supplementary Figure 2: The figure legend for panel C probably needs rephrasing. Should it read "C. (left) Gene Set Enrichment Analysis (GSEA) of differential expression [...] (right) GSEA of hallmarks of cancer pathways [...]?"

4. The Web app is currently unavailable. Since it is a central piece of the paper that enables the readers to easily access the results, it would be important to ensure its stability at the time of publication of the manuscript.

5. The authors have provided, in a GitHub repository, the code that performs the core analyses, including purity correction. However, perhaps because of storage space restrictions, they have not included the gene expression matrices that are required for running the script (i.e. the files "CCLE_normalized_expression.txt" and "< cancer > _ _normalized_expression.txt"). I suspect that these matrices will be of great interest to the readers not just for the purpose of replicating the results, but also because of the batch correction and normalization that was performed. Would it be possible to make these matrices available as compressed Supplementary Data Files as part of

the paper? I understand that the size of each file may be hundreds of megabytes – perhaps the editor can kindly provide alternative suggestions to the authors. An alternative would be to provide the a code that can regenerate the normalized matrices from transcript counts that are downloaded from Google Cloud Pilot RNA-Sequencing OSF repository.
es.

Reviewers' comments:

Reviewer #1 (Remarks to the Author):

I think the addition of the subtype information is very useful. For the subtype comparison versus your all tumor comparison, did you see major differences in correlation by adding in the tumor subtype information?

We thank the reviewer for their comment. For some tumor types such as PAAD, the cell line rankings are similar in the individual subtype comparisons compared to the all tumor comparison. In other tumor types such as BRCA, the rankings changed substantially in the individual subtype comparisons compared to the all tumor comparison. We have added the PAAD individual subtype comparisons as Supplementary Figure 4 and we have also updated the web application (<http://comphealth.ucsf.edu/TCGA110/>) to include the individual subtype comparisons for the other tumor types.

We have added the following to the results on page 8:

“Correlations between the pancreatic cell lines and the primary tumors in each individual subtype were also calculated (Supplementary Figure 4). The rankings of the pancreatic cell lines compared to the primary tumors in the individual subtypes were similar to the rankings of the pancreatic cell lines compared to all of the pancreatic primary tumors, suggesting that global differences between the samples outweigh the subtype specific differences for PAAD.”

The purity correction using an updated sets of purity seems to have improved the analysis. Supp table 4 shouldn't it also have a ranking measurement more than just an FDR score

We thank the reviewer for their comment. We have added the cell line rankings to the supplementary table (now called Supplementary Table 7). The ranks are shown for each subtype comparison, which is described in the table.

Supplementary table 5 show the individual pairwise correlations, but it seems like it also needs the correlation to the tumor type or tumor subtype as a whole to be easily usable by other users.

We thank the reviewer for their comment. We have made individual tumor type matrixes available for download in the online web application (<http://comphealth.ucsf.edu/TCGA110/>).

Having looked at the supplement table 6 (top 500 genes and GO pathways, different from the table number listed in the response to reviewers), I'm now a little concerned about the genes used for correlation and the 500 genes selected for each tumor type. For example, in BRCA the top Biological Processes are extracellular matrix related which is not tumor-related and in the 500 genes, I'm not seeing standard well known markers like ESR1, ERBB2, etc.

We thank the reviewer for their comment. We chose to compare the expression profiles of the cell lines and primary tumor samples using the 5,000 most variable genes in order to examine the

global differences between the samples using an unbiased approach rather than focusing on known cancer genes.

That being said, only the top 500 most variable genes (out of 5,000) are shown in the supplementary table (now called Supplementary Table 3). This cutoff was applied to have a reasonable number of genes for the gene ontology analysis. We have tried several approaches of selecting the top 500 genes (based on IQR and change in rank between cell lines and tumor samples) but that hasn't affected the pathway enrichment results. Genes such as ESR1, while missing from the top 500 gene list for BRCA, were included among the 5,000 most variable genes used in the correlation analysis for BRCA.

Reviewer #2 (Remarks to the Author):

I thank the authors for their revised manuscript. My major concerns have been addressed – most importantly, the new comparison of OV cell lines to a previous ranking based on an orthogonal approach provides evidence that the authors expression-based approach is in fact able to pick cell lines that reflect the characteristics of their intended tumours. Also, the suggestion by reviewer #1 to perform separate analysis of subtypes was excellent, and it appears to me that this other major addition has been done properly by the authors (method is appropriate and cross-validation seems to support the utility of the approach). Other technical concerns such as batch effect correction also seem to have been addressed. I have a few minor comments that I hope would further improve the manuscript:

1. As I mentioned, the authors have now included a comparison of their rankings for OV cell lines to a previous publication, but have only mentioned the Spearman correlation and its p-value. Can the authors kindly provide a supplementary table with the information for these cell lines, their rankings in the previous publication that was cited, and their expression correlation based on the author's approach?

We thank the reviewer for their suggestion and have added Supplementary Table 5 which includes our OV cell line correlations, the cell line suitability score in the Domcke et al publication, and the respective cell line rankings from each approach.

2. Figure 5: The panel labels need to be corrected (panels should be A-D).

We thank the reviewer for pointing out this oversight. Figure 5 has been corrected with the proper labels for each panel.

3. Supplementary Figure 2: The figure legend for panel C probably needs rephrasing. Should it read “C. (left) Gene Set Enrichment Analysis (GSEA) of differential expression [...] (right) GSEA of hallmarks of cancer pathways [...]”?

We thank the reviewer for their comment and apologize for the poor phrasing – the legend for Supplementary Figure 2 has been changed to the following:

“C. (left) Gene Set Enrichment Analysis (GSEA) of differential expression results without purity as a covariate between primary tumor samples and cell lines in hallmark gene sets from MSigDB. NES are shown for pathways with FDR < 5%. Before adjusting for tumor purity, immune related pathways are strongly enriched in primary tumor samples. (right) Gene Set Enrichment Analysis (GSEA) of differential expression results without purity as a covariate between primary tumor samples and cell lines in hallmarks of cancer pathways. NES are shown for pathways with FDR < 5%.”

4. The Web app is currently unavailable. Since it is a central piece of the paper that enables the readers to easily access the results, it would be important to ensure its stability at the time of publication of the manuscript.

We apologize for any issues encountered while accessing the web application. We have since made sure that the web application is available online.

5. The authors have provided, in a GitHub repository, the code that performs the core analyses, including purity correction. However, perhaps because of storage space restrictions, they have not included the gene expression matrices that are required for running the script (i.e. the files “CCLE_normalized_expression.txt” and “<cancer>__normalized_expression.txt”). I suspect that these matrices will be of great interest to the readers not just for the purpose of replicating the results, but also because of the batch correction and normalization that was performed. Would it be possible to make these matrices available as compressed Supplementary Data Files as part of the paper? I understand that the size of each file may be hundreds of megabytes – perhaps the editor can kindly provide alternative suggestions to the authors. An alternative would be to provide the a code that can regenerate the normalized matrices from transcript counts that are downloaded from Google Cloud Pilot RNA-Sequencing OSF repository.

We thank the reviewer for their comments and apologize for not providing the normalized gene expression matrixes earlier – as the reviewer had surmised, the size of the files exceeded the Github file size limit and failed to upload. We have since uploaded the gene expression matrices to Synapse (Synapse ID: syn18685536, <https://www.synapse.org/#!/Synapse:syn18685536/files/>) so users will have access to the normalized expression data for each tumor type.

REVIEWERS' COMMENTS:

Reviewer #1 (Remarks to the Author):

The additions have improved the manuscript and I have no major additional issues.

Reviewer #2 (Remarks to the Author):

All comments are addressed.